# Diabetes-Related Lower Extremity Amputations in Romania: Patterns and Changes between 2015 and 2019

**DOI:** 10.3390/ijerph20010557

**Published:** 2022-12-29

**Authors:** Horaţiu Coman, Bogdan Stancu, Norina A. Gâvan, Frank L. Bowling, Laura Podariu, Cosmina I. Bondor, Gabriela Radulian

**Affiliations:** 1Vascular Surgery Clinic, Cluj County Emergency Hospital, 400347 Cluj-Napoca, Romania; 2Second Department of Surgery, “Iuliu Haţieganu” University of Medicine and Pharmacy, 400006 Cluj-Napoca, Romania; 3Wörwag Pharma Romania SRL, 400267 Cluj-Napoca, Romania; 4Developmental Biomedicine Research Group, The University of Manchester, Manchester M13 9PL, UK; 5Department of Vascular Surgery and Reconstructive Microsurgery, “Victor Babeş” University of Medicine and Pharmacy, 300041 Timisoara, Romania; 6“Nicolae Stăncioiu” Heart Institute, 400001 Cluj-Napoca, Romania; 7Department of Medical Informatics and Biostatistics, “Iuliu Haţieganu” University of Medicine and Pharmacy, 400349 Cluj-Napoca, Romania; 8Department of Internal Medicine, “Carol Davila” University of Medicine and Pharmacy, 050474 Bucureşti, Romania; 9“Prof. Dr. Nicolae Paulescu” National Institute for Diabetes, Nutrition and Metabolic Diseases, 030167 Bucuresti, Romania

**Keywords:** diabetes mellitus, lower extremity amputation, incidence, nationwide study

## Abstract

Lower extremity amputations (LEAs) are a feared complication of diabetes mellitus (DM). Here we evaluated the recent trends in DM-related LEAs in Romania. We collected data from a national database regarding minor and major LEAs performed between 2015 and 2019 in patients with DM admitted to a public hospital. Absolute numbers of LEAs were presented by year, diabetes type, sex and age; incidence rates of LEAs in the general population were also calculated. Over the study period, 40,499 LEAs were recorded nationwide (83.16% in persons with type 2 DM [T2DM]); on average, the number of LEAs increased by 5.7%/year. This trend was driven by an increased number of LEAs in patients with T2DM; in patients with type 1 DM (T1DM), LEAs decreased over the study period. In patients with T2DM, the increase in minor LEAs was more pronounced than that in major LEAs. The overall number of LEAs showed an increasing trend with age (r = 0.72), which was most pronounced in patients aged ≥70 years. Men had a higher frequency of LEAs than women, regardless of DM type. These data support renewed efforts to prevent and decrease the burden of amputations among patients with DM.

## 1. Introduction

Diabetes mellitus (DM) represents a considerable burden: according to the International Diabetes Federation, in 2021 there were over 537 million adults living with DM worldwide, and this number is projected to rise to 783 million by 2045 [1]. In Europe, there are 61 million people with DM, and by 2045, an increase of 13% is predicted, which would amount to 69 million people [2]. In Romania, DM affects approximately 1.2 million people and is responsible for about 24,000 deaths [3].

DM and its complications represent the most common cause of nontraumatic lower extremity amputations [4,5]. Diabetes-related ulcerations are associated with an increased risk of limb amputation and mortality, in addition to a lower quality of life [6,7]. Patients who undergo a diabetes-related amputation have poorer outcomes compared with their counterparts with no amputations: the mortality rate of 48% in the first year after an amputation increases to over 70% within 5 years [8]. A study conducted in the United States showed that patients with amputations were hospitalized for longer and incurred costs that were three to six times higher than patients with healed diabetic foot ulcers [9]. Outcomes of patients with diabetic foot ulcers and amputations are influenced by multiple barriers on the individual, interpersonal and community levels as well as by ethnicity and race [10,11].

A study published in 2018 revealed that patients with DM fear amputations more than death [12]. Therefore, one of the main goals when treating diabetic patients should be to avoid any type of amputation and subsequently to reduce mortality rates and impact on healthcare systems. Diabetic foot care measures and amputation-prevention strategies should be at the core of the standard medical care provided to patients with DM.

Over the last decade, there has been significant improvement in international strategies regarding diabetic foot treatment [13,14], as well as in vascular devices and techniques for limb salvage [15,16,17]. Regarding the effect of national strategies in diabetic foot care in Romania [6,18], a previous nationwide evaluation conducted between 2006 and 2010 has revealed relatively high amputation rates [19]. Vereşiu et al. [19] showed that over 4000 amputations/year were performed between 2006 and 2010, amounting to more than 24,000 amputations in the 5-year period covered by the study. The incidence of diabetes-related amputations/100,000 persons/year in the general population increased every year, from 18.15 in 2006 to 25.52 in 2010 [19].

In this study, we provide an update of the trends in diabetes-related minor and major lower extremity amputation rates in Romania for the 2015–2019 period.

## 2. Materials and Methods

We retrospectively analyzed data on all diabetes-related lower extremity amputations performed during admission to a public hospital in Romania, as previously described [19]. Amputations performed between 1 January 2015 and 31 December 2019 were included in this re-evaluation study.

Briefly, data were obtained from the National School for Public Health, Management and Health Education (Şcoala Naţională de Sănătate Publică, Management şi Perfecţionare) [20] and included the total number and the level of the amputations performed, the type of diabetes and the 10-year age bracket of the patients who underwent amputations. Relevant data were selected from the national database using the following criteria:-Presence of a primary or secondary diagnostic of DM at discharge (International Classification of Diseases 10 [ICD 10] codes E10, E11, E13 and E14).-A major or minor amputation procedure; major amputations included hip amputation (ICD 10 AM v.3 code 44370-00), amputation above or below the knee (44367-00, 44367-02) and ankle amputation through the malleoli of the tibia and fibula (44361-01), whereas minor amputations included ankle disarticulation (44361-00), midtarsal and transmetatarsal amputation (44364-00, 44364-01), toe amputations with/without metatarsal bone (44358-00, 44338-00) and toe disarticulation (90557-00).

Summaries were generated using descriptive statistics. Official population data published by the National Institute of Statistics [21,22,23,24,25] and the diabetes-related data published by the National Institute of Public Health [26] was used to calculate the crude incidence of diabetes-related amputations/100,000 persons/year for the general population and for patients with DM. The rate of minor to major amputations was also calculated.

The trend analysis was performed using linear regression. A linear equation was computed, and the Pearson coefficient of correlation was calculated. The level of statistical significance was set at 0.05.

Statistical analysis was performed with Microsoft Excel 2016.

## 3. Results

Between 2015 and 2019, a total of 40,499 diabetes-related amputations were recorded nationwide, 13.84% of which were in persons with type 1 DRG-coded DM (T1DM); 83.16% were in persons with type 2 DRG-coded DM (T2DM); 1.14% of amputations were in persons who had other DRG-coded types of diabetes; and the remaining 1.86% were in patients with nonspecified DRG-coded types of diabetes diagnosed at discharge. The yearly breakdown of the absolute number of diabetes-related lower extremity amputations, as well as the crude incidence of diabetes-related amputations in the general population and in persons with DM, is presented in Table 1. The numbers corresponding to the different amputation type for each study year are presented in the Appendix A.

Over the 5 study years, we found a mean (± standard deviation [SD]) of 8099.80 (±709.23) amputations/year. According to diabetes type, the mean (±SD) numbers of lower extremity amputations over the studied period were 1121.2 (±135.63) in patients with T1DM, 6735.4 (±838.82) in patients with T2DM, 92.6 (±12.05) in patients with other diabetes types and 150.6 (±8.26) in patients with nonspecified types of diabetes. Overall, the number of amputations increased over the 5-year period by a mean rate of 5.7%/year, amounting to a total increase of 24.7% in 2019 compared with 2015. This trend was driven by variations in the number of amputations in T2DM patients, which increased by 9.3%, 22.1%, 28.9% and 37.0% in 2016, 2017, 2018 and 2019, respectively, compared with the baseline (2015). In patients with T1DM, the number of amputations decreased over the study period and varied by −8.0%, −15.7%, −21.9% and −25.6% in 2016, 2017, 2018 and 2019, respectively, compared with 2015 (Figure 1).

To predict the number of amputations, the following equations can be used:y = 172,566 − 85x(1)
for patients with T1DM (r = −0.99, *p* = 0.001, r^2^ = 0.98) and
y = 527.7x − 1,057,635.5(2)
for patients with T2DM (r = 1.00, *p* < 0.001, r^2^ = 0.99), where x is the year for which we want to compute the prediction and y is the predicted number of amputations in patients with T1DM or T2DM. According to these equations, if the current trends continue, in patients with T1DM, the number of amputations will decrease to 441, corresponding to about one-half of the cases recorded in 2019. On the other hand, 10,957 amputations are predicted for the same year in patients with T2DM, corresponding to an increase of 41.8% compared with 2019.

When stratifying the findings according to the level of the amputation, the absolute number of minor and major amputations decreased similarly in patients with T1DM (Figure 2A), whereas in patients with T2DM, the increase in minor amputations was more pronounced than that in major amputations (Figure 2B). Over the 5 study years, the minor–major amputation ratio increased more in patients with T1DM compared with those with T2DM: 2.69 vs. 2.26 in 2015, 2.63 vs. 2.39 in 2016, 2.84 vs. 2.46 in 2017, 3.38 vs. 2.38 in 2018 and 3.34 vs. 2.57 in 2019.

Men had a higher frequency of lower extremity amputations than women, regardless of whether they had T1DM or T2DM (Figure 3). Variations in the numbers of amputations compared with the baseline (2015) followed a decreasing trend in patients with T1DM, both in women (−4.1%, −9.5%, −29.7% and −35.1% for 2016, 2017, 2018 and 2019, respectively) and in men (−9.6%, −18.3%, −18.5% and −21.5% for 2016, 2017, 2018 and 2019, respectively). Conversely, in patients with T2DM, this trend was increasing, in women (8.3%, 18.8%, 30.5% and 35.4% for 2016, 2017, 2018 and 2019, respectively) and men (9.7%, 23.3%, 28.3% and 37.6% for 2016, 2017, 2018 and 2019, respectively) alike. Between 2015 and 2019, the men–women ratio of overall amputations was 2.35, 2.22, 2.12, 2.73 and 2.85 in patients with T1DM, with a mean of 2.45, whereas in patients with T2DM, this ratio was 2.64, 2.67, 2.74, 2.59 and 2.68, with a mean of 2.66.

Overall, women had a lower number of amputations than men; while women with T1DM had a share of major amputations that was largely similar to that seen in men, in women with T2DM, the share of major amputations was up to 10% higher compared with men (Table 2). The risk of major amputations was also higher in women with T2DM compared with men, for all study years: 1.37 in 2015, 1.33 in 2016, 1.21 in 2017, 1.27 in 2018 and 1.23 in 2019. The ratio of minor-to-major amputations was lower in women with T1DM than in men: 2.12 vs. 3.00 in 2015, 2.56 vs. 2.67 in 2016, 2.53 vs. 3.01 in 2017, 3.22 vs. 3.45 in 2018 and 2.83 vs. 3.56 in 2019. The same was true for women and men with T2DM: 1.62 vs. 2.59 in 2015, 1.78 vs. 2.69 in 2016, 2.02 vs. 2.65 in 2017, 1.86 vs. 2.63 in 2018 and 2.08 vs. 2.80 in 2019.

In patients with T1DM, lower extremity amputations decreased over the study period more prominently in women than in men (Figure 4A); this was true for both minor and major amputations but was more evident for the latter category. Compared with the baseline (2015), the variations in the number of major amputations in women and men with T1DM were −16.0% vs. −1.3% for 2016, −20.0% vs. −18.3% for 2017, −48.0% vs. 26.6% for 2018 and −47.2% vs. −31.0% for 2019. In comparison, the variations in the number of minor amputations from the baseline (2015) for T1DM in women and men were 1.51% vs. −12.40% for 2016, −4.53% vs. −18.30 for 2017, −21.13% vs. −15.80% for 2018 and −29.43% vs. 18.30% for 2019.

In patients with T2DM, over the study period, women had a more pronounced increase in minor amputations than men, but major amputations increased more in men than in women (Figure 4B). Compared with the baseline (2015), the variations in the number of minor amputations from baseline (2015) for women and men with T2DM were 12.1% vs. 10.8% for 2016, 28.5% vs. 24.0% for 2017, 37.1% vs. 28.8% for 2018 and 47.9% vs. 40.5% for 2019. On the other hand, variations in the number of major amputations in women and men with T2DM were 2.2% vs. 6.9% for 2016, 3.2% vs. 21.4% for 2017, 19.6% vs. 27.1% for 2018 and 15.2% vs. 30.1% for 2019.

Age did not appear to influence the number of lower extremity amputations in patients with T1DM, as the decrease over time was similar in all age categories (r = 0.04). However, in patients with T2DM, the number of amputations showed an increasing trend with age (r = 0.72), with the most pronounced increase observed in patients with T2DM ≥70 years of age (Figure 5). The limited and variable number of amputations recorded in the <30 and the 30–39 age groups do not allow any relevant conclusions to be drawn.

Between 2015 and 2017, the incidence of amputations was highest in patients with T2DM who were aged 60–69 years. However, in this age group, amputations followed a decreasing trend, and starting with 2018, the highest incidence was seen in patients with T2DM who were aged ≥70 years (Table 3).

Overall, the share of major amputations (from all diabetes-related lower-extremity amputations) appears to increase with age, although a decreasing trend is seen over the study period (Table 4). This finding indicates that the overall increase in lower extremity amputations observed over the study period is driven by an increase in minor amputations. Some differences exist between the trends observed in women and men. In men, the share of major amputations remains relatively constant over the study period, with differences between −2.8% and −0.7% in the different age groups. In women ≥40 years of age, the share of major amputations decreased over the study years; this decrease was more pronounced in the older age groups, between −1.4% in women aged 40–49 years and −6.8% in women aged ≥70 years.

## 4. Discussion

This retrospective study that spans 5 years (2015–2019) and examines diabetes-related amputations in Romania aims to provide an update of the last evaluation of such data, published by Vereşiu et al. in 2010 [19].

Overall rates of diabetes-related lower extremity amputations have increased over the 5 years of our study. The absolute number of amputations recorded between 2015 and 2019 in Romania was 40,499, amounting to a mean (±SD) of 8099.80 (±709.23) amputations/year, compared with 4584.4 (±612.42) amputations/year reported by Vereşiu et al. [19]

This increase was driven by a rise in both minor and major amputations in patients with T2DM and especially in older individuals: in patients with T2DM, we found a total of 33,677 lower extremity amputations, compared with 15,330 amputations reported in this population between 2006 and 2010 [19].

Previously published data from the PREDATORR study showed that in Romania, the prevalence of DM is higher in men than in women, whereas prediabetes was more prevalent in women [18]. In line with these results, in our study, the number of diabetes-related amputations was higher in men than in women, regardless of whether the patients had T1DM or T2DM. We found that men had a 2.75 times higher risk of amputation compared with women; our results are similar to those of an Austrian study conducted between 2014 and 2017, which showed a 2.53 times higher amputation rate in men compared with women [27].

The incidence of diabetes-related lower extremity amputations in the general population varied between 17.62 and 21.23 amputations/100,000 persons/year in women, while in men, it varied between 47.53 and 60.51 amputations/100,000 persons/year. Lower incidences of diabetes-related minor lower extremity amputations in women compared with men also appeared to be a common finding for many of the countries included in the study conducted by Hughes et al. [28]. For example, in 2017, the highest amputation rates were observed in Australia (34.8/100,000 persons in women and 41.9/100,000 persons in men); the lowest amputation rate in women (19.4/100,000 persons) was recorded in Italy; and the lowest rate in men (31.2/100,000 persons) was observed in the Netherlands. However, in the United States, the country with the lowest rates of minor lower extremity amputations in both sexes, the rate of amputations was 12.8/100,000 in women and 12.4/100,000 in men [28].

Although there was an overall increase in lower extremity amputations over the study period, in patients with T1DM, the number of amputations (both minor and major) decreased, especially in women. In patients with T2DM, the number of major lower extremity amputations, as well as the share of major amputations from all amputations, increased with age in all study years. Over time, however, the share of major amputations decreased from 2015 to 2019 in all age groups ≥40 years, with a more prominent decrease observed in women. This finding allows us to conclude that the overall increase in lower extremity amputations over time reported here is driven by an increase in minor amputations, which is more marked in women than in men. A similar finding has been reported by Kröger et al. [29], who analyzed official lower extremity amputation data from all hospitals in Germany between 2005 and 2014. They found that while the rate of major amputations decreased by 30.6% over the study period, the rate of minor amputations increased by 25.4% [29].

There could be several factors that led to this diverging evolution of amputations in patients with T1DM and T2DM, respectively. Patients with T1DM are usually younger and have fewer comorbidities than patients with T2DM; as symptom onset is faster with T1DM, these patients also tend to seek medical care earlier than those with T2DM [30]. Visiting a physician earlier in the course of the disease might also be influenced by the fact that younger individuals generally have access to more-diversified information resources than older individuals do [31]. As previously demonstrated, delay in seeking medical care can increase the risk of amputations in patients with DM [32].

In our study, the highest incidence of lower extremity amputations (146.35/100,000 persons/year) was observed in 2019, in patients with T2DM aged ≥70 years. These findings are in line with those of a study conducted in Austria between 2014 and 2017, where 83% of the 2165 patients who had major lower extremity amputations had T2DM, and their mean age was 73.0 ± 11.3 years [27]. The highest incidence of amputations shifted from the 60–69-year-old age group (between 2015 and 2017) to the oldest age group of patients, ≥70 years of age, in 2018; this could be explained either by the increased survival of patients with T2DM or by an improvement in diabetic foot care that delays lower extremity amputations.

Retrospective data on major lower extremity amputations in Trinidad and Tobago between 2010 and 2016 showed that 79% of patients who had an amputation presented more than 7 days after the onset of symptoms, and sepsis was the most common indication for amputation (in 71.5% of cases) [33]. In Romania, most patients sought medical care within 1 to 6 months after the onset of symptoms of DM or its complications. The risk of gangrene and amputations became significantly higher in those who sought medical care after 1 year from symptoms onset and more than doubled in those who sought medical care after 2 years from symptoms onset [32]. The presence of diabetic foot complications, such as ulcers and amputations, has also been shown to increase the cost of hospitalization by more than 40%, compared with patients without complications [6].

Recently, interest has increased in exploring the benefits of involving a multidisciplinary team in the prevention and treatment of diabetic foot complications [13]. There are multiple studies that report the positive impact of a comprehensive, multidisciplinary approach to diabetic limb salvage. A reduction in the incidence of overall amputations, improved wound healing, reduced length of hospital stay and a decrease in recurrent ulcerations have all been associated with a multidisciplinary team [13,14]. Therefore, there is reason to believe that patient care would be improved by setting up diabetic foot teams [14].

This study has several limitations related to the retrospective method of data acquisition. Our analysis included only cases of standalone lower extremity amputations because of the lack of reliable data on repeated amputations in the same individual. We have no follow-up information about the patients who had amputations or long-term mortality data. Furthermore, no details regarding patient- and healthcare-related factors such as the duration of diabetes, indications for amputations, ethnic background, social index or clinical and laboratory parameters were available. Finally, data limitations precluded the examination of limb salvage efforts such as debridement or revascularization procedures in relation to the risk of amputation. A prospective study designed with awareness of the complex problem posed by amputations should avert these limitations in the future.

While our study was not designed to assess the impact of various preventive strategies in lowering the number of diabetes-related lower extremity amputations, we believe that efforts directed towards improving the screening process, providing continued medical educational to both patients and healthcare providers and increasing addressability will decrease the burden of amputations on patients and healthcare systems alike.

## 5. Conclusions

The number of diabetes-related lower extremity amputations in Romania has increased since the last evaluation conducted between 2006 and 2010. An increasing trend in the number of amputations was also seen over the study period (2015–2019), which was driven by an increase in minor amputations in patients with T2DM.

Age did not appear to influence the number of lower extremity amputations in patients with T1DM, as the decrease over time was similar in all age categories. Conversely, in patients with T2DM, the number of amputations increased with age, with the most pronounced increase observed in patients with T2DM who were aged ≥70 years.

Although the risk of major amputations was higher in women with T2DM compared with men, major amputations increased more in men than in women for all study years; women had a more pronounced increase in minor amputations than men.

In patients with T1DM, lower extremity amputations decreased over the study period.

These data support renewed efforts to prevent and decrease the burden of amputations among patients with DM.

## Figures and Tables

**Figure 1 ijerph-20-00557-f001:**
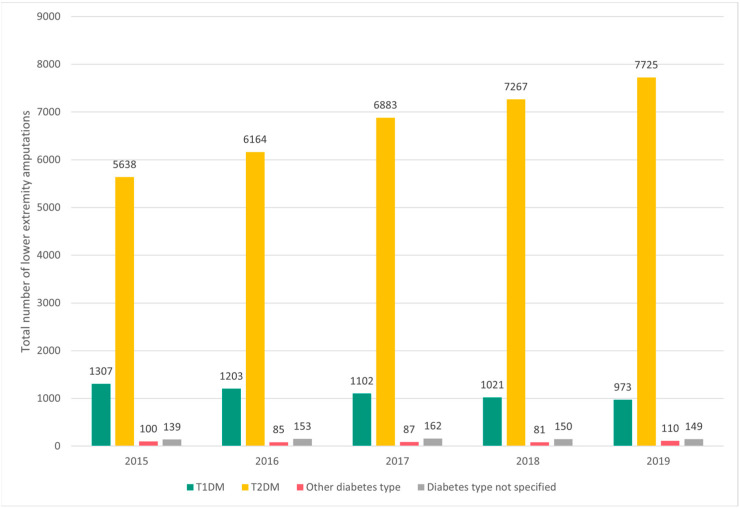
Total number of nontraumatic lower extremity amputations per year, according to diabetes type. T1DM, type 1 diabetes mellitus; T2DM, type 2 diabetes mellitus.

**Figure 2 ijerph-20-00557-f002:**
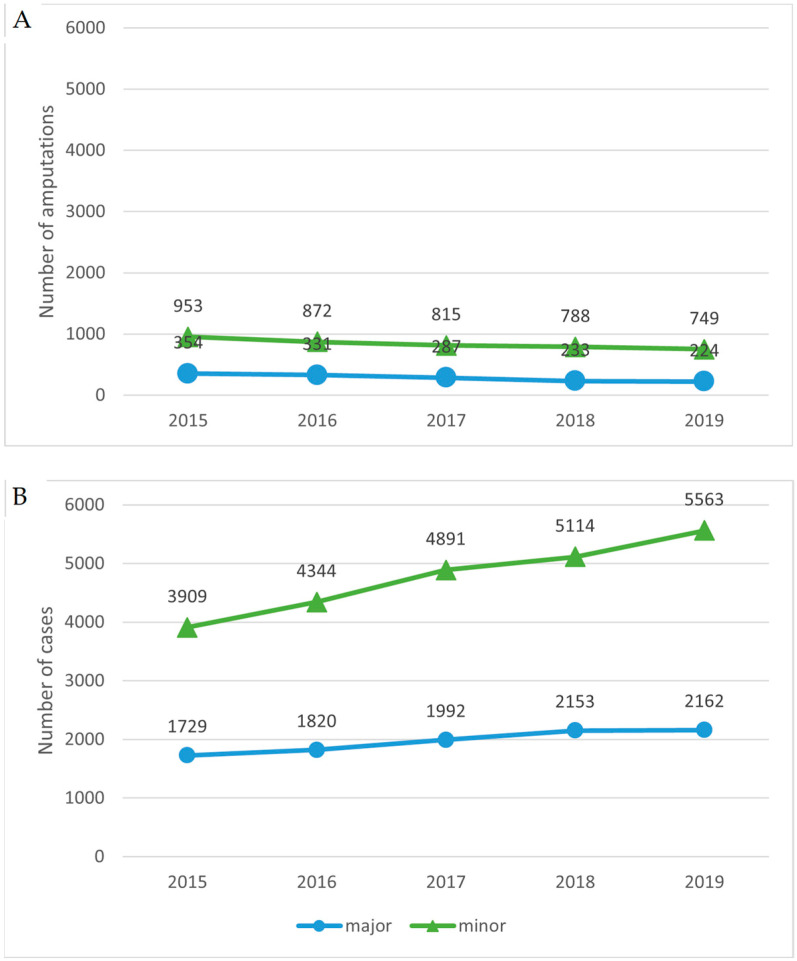
Trends in the number of major and minor lower extremity amputations in patients with T1DM (**A**) and T2DM (**B**). T1DM, type 1 diabetes mellitus; T2DM, type 2 diabetes mellitus.

**Figure 3 ijerph-20-00557-f003:**
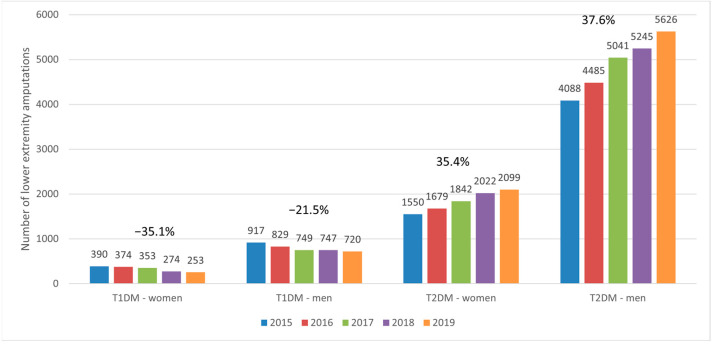
Trends in the number of lower extremity amputations in patients with T1DM and T2DM, according to sex. T1DM, type 1 diabetes mellitus; T2DM, type 2 diabetes mellitus.

**Figure 4 ijerph-20-00557-f004:**
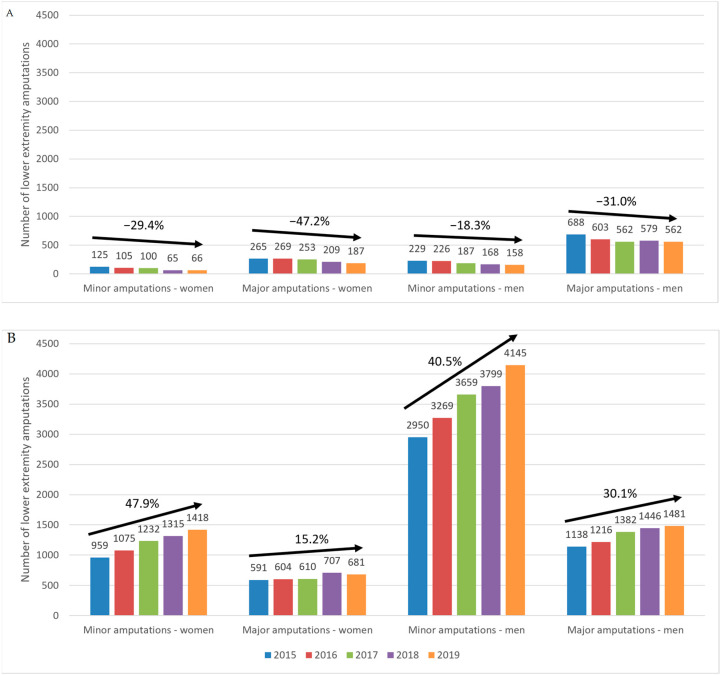
Trends in the number of lower extremity amputations in patients with T1DM (**A**) and T2DM (**B**) according to sex and level of amputation. T1DM, type 1 diabetes mellitus; T2DM, type 2 diabetes mellitus.

**Figure 5 ijerph-20-00557-f005:**
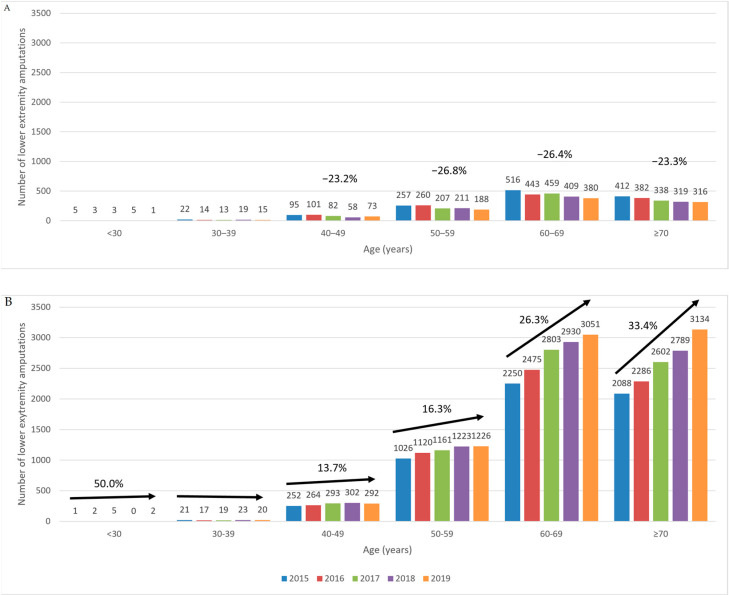
Trends in the numbers of lower extremity amputations in patients with T1DM (**A**) and T2DM (**B**) diabetes, according to age. T1DM, type 1 diabetes mellitus; T2DM, type 2 diabetes mellitus.

**Table 1 ijerph-20-00557-t001:** Absolute number, crude incidence and level of amputations by year.

	2015 [22]	2016 [22]	2017 [23]	2018 [24]	2019 [25]	Total
Amputations	7184	7605	8234	8519	8957	40,499
Incidence of DM-related amputations in the general population (/100,000 persons/year)	32.22	34.19	37.05	38.38	40.40	NA
Incidence of DM-related amputations in patients with DM (/100,000 persons/year) [26]	719.63	746.43	777.89	733.56	762.82	NA
Major amputations (% of total amputations)	2150 (29.9)	2227 (29.3)	2354 (28.6)	2452 (28.8)	2455 (27.4)	11,638 (28.7)
Minor amputations (% of total amputations)	5034 (70.1)	5378 (70.7)	5880 (71.4)	6067 (71.2)	6502 (72.6)	28,861 (71.3)

Data on the population of Romania for each study year was extracted from the official data published by the National Institute of Statistics (references indicated in the table heading for each year). Data on the population with DM for each study year was extracted from the official data published by the National Institute of Public Health (reference indicated in the corresponding row). DM, diabetes mellitus; NA, not applicable.

**Table 2 ijerph-20-00557-t002:** Absolute number, crude incidence and level of amputations according to year, type of diabetes and sex.

	2015 [22]	2016 [22]	2017 [23]	2018 [24]	2019 [25]	Total
Men–women ratio of risk of amputations in the general population	2.70	2.73	2.75	2.74	2.85	2.75 *
Incidence of diabetes-related amputations in women from the general population (/100,000 women/year)	17.62	18.52	20.00	20.77	21.23	NA
Incidence of diabetes-related amputations in men from the general population (/100,000 men/year)	47.53	50.62	54.94	56.87	60.51	NA
Amputations in women with T1DM	390	374	353	274	253	1644
Major amputations (%)	125 (32.1)	105 (28.1)	100 (28.3)	65 (23.7)	66 (26.1)	461 (28.0)
Minor amputations (%)	265 (67.9)	269 (71.9)	253 (71.7)	209 (76.3)	187 (73.9)	1183 (72.0)
Amputations in men with T1DM	917	829	749	747	720	3962
Major amputations (%)	229 (25.0)	226 (27.3)	187 (25.0)	168 (22.5)	158 (21.9)	968 (24.4)
Minor amputations (%)	688 (75.0)	603 (72.7)	562 (75.0)	579 (77.5)	562 (78.1)	2994 (75.6)
Amputations in women with T2DM	1550	1679	1842	2022	2099	9192
Major amputations (%)	591 (38.1)	604 (36.0)	610 (33.1)	707 (35.0)	681 (32.4)	3193 (34.7)
Minor amputations (%)	959 (61.9)	1075 (64.0)	1232 (66.9)	1315 (65.0)	1418 (67.6)	5999 (65.3)
Amputations in men with T2DM	4088	4485	5041	5245	5626	24,485
Major amputations (%)	1138 (27.8)	1216 (27.1)	1382 (27.4)	1446 (27.6)	1481 (26.3)	6663 (27.2)
Minor amputations (%)	2950 (72.2)	3269 (72.9)	3659 (72.6)	3799 (72.4)	4145 (73.7)	17,822 (72.8)

Data source on population of Romania for every year of the study period was the official data published by the National Institute of Statistics (references indicated in the table for each year). * risk computed using the average number of the population over the 5 study years for both women and men.

**Table 3 ijerph-20-00557-t003:** Incidence of diabetes-related lower extremity amputations/100,000 persons/year, according to age in the general population.

Age	2015 [22]	2016 [22]	2017 [23]	2018 [24]	2019 [25]
<30	0.08	0.07	0.12	0.08	0.04
30–39	1.25	0.95	1.02	1.22	1.02
40–49	10.12	10.13	10.50	9.83	10.04
50–59	47.93	53.51	54.50	54.58	51.61
60–69	118.47	118.83	128.62	129.33	130.85
≥70	108.43	116.47	127.36	132.59	146.35

Data source on population of Romania for every year of the study period was the official data published by the National Institute of Statistics (references indicated in the table for each year).

**Table 4 ijerph-20-00557-t004:** Major lower extremity amputations by years, according to sex and age (absolute number [percentage]).

	Age	2015	2016	2017	2018	2019
Total	<30	1 (16.7)	0 (0.0)	0 (0.0)	1 (16.7)	0 (0.0)
	30–39	5 (11.1)	6 (17.6)	6 (16.7)	9 (20.9)	6 (16.7)
	40–49	77 (21.4)	65 (17.4)	81 (20.1)	80 (21.6)	76 (20.5)
	50–59	322 (24.2)	336 (23.5)	296 (21)	305 (20.6)	309 (21.0)
	60–69	807 (28.3)	835 (27.8)	906 (27.1)	958 (27.8)	922 (26.1)
	≥70	938 (36.3)	985 (35.8)	1065 (35.1)	1099 (34.6)	1142 (32.2)
Women *	<30	0 (0.0)	0 (0.0)	0 (0.0)	0 (0.0)	0 (0.0)
	30–39	1 (9.1)	3 (33.3)	3 (15.8)	2 (13.3)	1 (11.1)
	40–49	19 (25.7)	14 (16.7)	17 (20.0)	10 (14.9)	17 (24.3)
	50–59	74 (27.8)	74 (26.8)	69 (25.6)	67 (25.3)	52 (24.4)
	60–69	239 (35.2)	209 (28.5)	241 (29.7)	248 (31.6)	231 (29.2)
	≥70	408 (41.7)	429 (42.7)	411 (37.8)	468 (38.2)	463 (34.9)
Men	<30	1 (33.3)	0 (0.0)	0 (0.0)	1 (20.0)	0 (0.0)
	30–39	4 (11.8)	3 (12.0)	3 (17.6)	7 (25.0)	5 (18.5)
	40–49	58 (20.3)	51 (17.6)	64 (20.2)	70 (23.1)	59 (19.6)
	50–59	248 (23.3)	262 (22.7)	227 (20.0)	238 (19.6)	257 (20.4)
	60–69	568 (26.1)	626 (27.5)	665 (26.2)	710 (26.7)	691 (25.2)
	≥70	530 (33.0)	556 (31.8)	654 (33.6)	631 (32.4)	679 (30.6)

* Proportions of major amputations that are higher in women compared with the corresponding proportion in men are marked in grey.

## Data Availability

Restrictions apply to the availability of the data obtained from the National School for Public Health, Management and Health Education. These data are available from the authors with the permission of the National School for Public Health, Management and Health Education.

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
