# Peer review of "Diabetes-Related Lower Extremity Amputations in Romania: Patterns and Changes between 2015 and 2019"

_ijerph, 2022, doi:10.3390/ijerph20010557_

Round 1

Reviewer 1 Report

Lower extremity amputations (LEAs) are a feared complication of diabetes mellitus (DM). Congratulations. After reviewing a total of 40,499 cases of Lower extremity amputations from 2015 to 2019, the article concludes that the number of amputations due to type 2 DM is gradually increasing, and Amputations due to type 1 DM tended to decrease. But I hope that the author can take a closer look at the reasons for the formation of such statistical data, and also explain why such data can prevent and decrease the burden of amputations among patients with DM.

Reviewer 2 Report

This is a very important article that continues to highlight the global problem that is amputation as a result of diabetes mellitus. This article clearly shows amputation trends in Romania which, as in other European countries, has increased; an alarming trend that requires immediate address by the competent authorities on national levels.

Some small suggestions to authors:

line 51-52: perhaps the main goals to avoid amputation are not only patient fear, but other factors such as mortality rates and impact on patient and health systems

Line 60: has revealed high amputation rates - perhaps here authors could provide some simple statistics to demonstrate what these high amputation rates were, so that the reader would not have to research for the referenced paper

Line 90: which version of Microsoft Excel?

Conclusion: I think a two sentence conclusion is too dry. I would suggest that this is beefed up, to include also what the actual findings were.

I would also suggest to the authors that perhaps it is time to look at the effect that the Covid-19 pandemic had on these rates.

Reviewer 3 Report

This is a descriptive population-based study from a national registry that has a significantly large number of patients. It showed an increased incidence of diabetes-related amputations despite advances in medical care.  

Author Response

Following comments received from the reviewers, we amended the manuscript by adding more background data with supporting references, clarifications to the Methods section, additional discussion points, and a more detailed conclusion. The changes made are highlighted with track changes in the revised version of the manuscript.

Round 2

Reviewer 1 Report

the manuscript has been sufficiently improved to warrant publication in IJERPH